# Study on the Performance of Ag-Cu Bimetal SERS Substrate

**Xiaolong Song [1], Xiaoya Yan [2], Na Li [1], Lin Shen [3],\* and Mingli Wang [1],\***

1. State Key Laboratory of Materials Science & Technology and Key Laboratory for Microstructural Material Physics of Hebei Province, School of Science, Yanshan University, Qinhuangdao 066004, China
2. Institute of Modern Optics, School of Physics, Harbin Institute of Technology, Harbin 150001, China
3. College of Liren, Yanshan University, Qinhuangdao 066004, China
\* Correspondence: shenlin@ysu.edu.cn (L.S.); wml@ysu.edu.cn (M.W.)

**Abstract:** SERS has become a powerful trace detection technology, but its practical application is often limited by the fixed optical properties of cast metals (Au, Ag and Cu). In this paper, the bimetallic nanostructures prepared by changing the metal content ratio can regulate the different optical responses of the substrate. In addition, it was found that the scale of moth wings (MW) with 3D grating-like uniform nanoarrays using bioscaffold can provide a consistently high-density 'hot spot' for the as-prepared plasmonic substrate. Here, two different methods (i) co-sputtered with different times and (ii) sputtered with sequentially alternating to form a stratified structure on the MW were employed for the fabrication of SERS-active substrates, and they were named AgCu@MW and Ag@Cu@MW, respectively. The comparison SERS measurement results of the above two substrates by using probe molecules R6G and MB show that the stratified structure of the Ag@Cu@MW-3 substrate has higher SERS performance. This strategy can not only prepare plasma substrates with on-demand optical responses but also provide new ideas for the preparation of conventional biomimetic nanomaterials.

**Keywords:** surface-enhanced Raman spectroscopy; moth wing; AgCu@MW; Ag@Cu@MW

## 1. Introduction

Surface-enhanced Raman spectroscopy (SERS) is an important spectroscopic analytical technique that has the potential of sensitivity single-molecule detection and has successfully employed as a useful tool for the identification of chemical and biological species [1–3].The extremely strong enhancement of SERS signal is primarily caused by the local electromagnetic enhancement mechanism that generated from the localized surface plasmons (LSP) in the plasmonic surfaces of metal nanostructures [4,5]. In addition, local surface plasmon resonance (LSPR) can generate an amplified electromagnetic field when molecules are adsorbed on the plasmonic surfaces of metal nanostructures [6]. In SERS, the detection of ultra-low concentration molecule, especially single-molecules, the distribution of nanogap ('hot spot') sites that can generate a high electric field on a plasmonic surface play a vital role [7,8]. It is well-known that only a small number of sites such as particle gap [9,10], nanopore [11,12] and sharp corner or tip [13,14] on plasmonic surfaces have an especially strong Raman signal [15]. Therefore, in order to obtain as many sites as possible where the adsorbed probe molecules overlap with the hot spots to achieve ultra-low concentration detection, it is essential to prepare plasmonic nanostructures with rough surfaces. Coinage metals (Au, Ag and Cu) have been extensively explored as the substrate source in the field of SERS due to their LSPR and surface plasmon polarization (SPP) characteristics of free electrons in metals in the visible range [16,17]. The optical response of this pure metallic nanostructure is fine-tuned by changing its size, shape and distribution, but the intrinsic properties of pure metal's dielectric function limit its application [18]. One pathway to adjust the dielectric function of pure metal materials for a stronger optical response is to shift the new exploration of these functional materials to more complex structures, such as

bimetallic nanostructures [19,20]. Compared with pure metallic materials, alloy or core–shell structures, plasmonic materials are considered to be able to synergistically improve the physical and chemical properties of the components due to the unique synergistic effects between different combinations of metal [21,22]. For example, previous studies of bimetallic CuAg have suggest that CuAg core–shell plasmonic substrates have excellent thermal stability [23], antioxidation [24] and chemical stability [25], and CuAg alloy plasmonic substrates have excellent strength [26], antibacterial activity [27] and reproducibility [11]. Therefore, varying the combinations of bimetal to control the optical response of plasmonic substrates is potentially very useful.

As we all know, the key to realizing a SERS practical application is to fabricate a 3D nanoarray SERS-active substrate with strong signal enhancement, superb sensitivity, excellent uniformity and wonderful reproducibility by a simple and cost-effective method. To date, various materials SERS substrates with controllable shapes, sizes and spacing through different nanofabrication techniques have been explored [28,29]. Among these plasmonic nanostructured substrates, 3D nanostructures are the most attractive and more suitable for practical applications of SERS, because 3D substrates provide high-density 'hot spots' and binding sites for undetermined molecules within the laser illumination area [30]. Unfortunately, although these existing 3D nanostructures may have extremely high SERS sensitivity, their practical applications are often limited due to their complex preparations, high costs and unstable structures [31,32]. In recent years, biological scaffolds have become frequently used templates for preparing SERS substrates, which not only tremendously simplify the preparation process but also reduce the cost because of the formation of a series of 3D, regular and suitable-size nanostructure arrays on the surfaces of their wings after a long period of evolution [33,34]. Therefore, it is of great significance to introduce well-developed prepared methods into these naturally designed 3D biological scaffolds.

In this paper, we present a simple and efficient method base on the biological scaffolds of moth wings to fabricate large-scale and 3D regular AgCu@MW and Ag@Cu@MW substrates. The rib gap in AgCu@MW can be modified by adjusting the co-sputtering time, and the sputtering layer of Ag@Cu@MW can be adjusted using a multi-target DC magnetron sputtering system. The screening and sensitivity analyses of the AgCu@MW and Ag@Cu@MW substrates have been completed by using R6G molecules, and the reproducibility and homogeneity of the AgCu@MW-20 and Ag@Cu@MW-3 substrates have been compared using MB molecules. Obviously, the as-prepared Ag@Cu@MW-3 substrate provides the optimum SERS activity, which includes a Raman intensity of R6G molecules at 1363 cm$^{-1}$, about eight times higher than that of the AgCu@MW-20 substrate; the sensitivity can be increased by two orders of magnitude compared with the AgCu@MW-20 substrate; the Ag@Cu@MW-3 substrate has a lower RSD value of less than 15%. Moreover, a comparison of the SERS performances between co-sputtering and sequential sputtering substrates can provide insights into the future designs and applications of SERS substrates.

## 2. Experimental

### 2.1. Materials and Instruments

Moth wings (MW) were purchased from Beijing Jiaying Grand Life Sciences Co., Ltd. (Beijing, China) Cu target material (99.99%) and Ag target material (99.99%) were purchased from Nanchang Material Technology Co., Ltd (Nanchang, China). R6G ($C_{28}H_{31}N_2O_3Cl$) and ethyl alcohol ($C_2H_6O$) were purchased from J&K scientific LTD (Beijing China). MB ($C_{16}H_{18}ClN_3S$) was purchased from Tianjin Zhiyuan Chemical Reagent Co., Ltd. (Tianjin China). Deionized water (DI) used in the entire experiment was purified in the Key Laboratory for Microstructural Material Physics of Hebei Province. The MW templates were decorated with CuAg nanoparticles by co-sputtering and sequentially alternating fashions with the High-Vacuum Multi-Target Magnetron Sputtering Equipment (JGP450). SERS and Raman measurements were performed using a Renishaw InVia Raman Microscope (Wotton-under-Edge, UK). The morphology of the MW template, AgCu@MW and Ag@Cu@MW

substrates were analyzed with the Field Emission scanning electron microscope (FE-SEM, Hitachi S-4800 II, Hitachi Ltd., Tokyo, Japan).

### 2.2. Preparation of SERS-Active Substrates

A series of SERS substrates decorated with Ag and Cu nanoparticles on MWs were prepared by the multi-target magnetron sputtering system: (i) co-sputtering (AgCu) by adjusting the sputtering time and (ii) sequentially alternating sputtering (Ag@Cu) with the total sputtering time maintain invariant. The fabrication process is shown in Figure 1. In the case of sequentially alternating depositions, the numbers of the alternating cycles were 1, 2 and 3, containing a pair of Ag and Cu layers in each cycle, and the substrate was maintained in the magnetron sputtering device to avoid contamination. Moreover, Ag terminates the surface layer in the alternating sputtering. In order to acquire the optimum SERS enhancement performance of the substrates, Cu was decorated by the direct current sputtering target, and Ag was decorated by the radio frequency sputtering target. In addition, the vacuum was less than $10^{-3}$ pa under the two sputtering modes. The co-sputtering time was adjusted from 10 to 25 min with a fixed interval of 5 min, and the corresponding as-deposited substrates were denoted as AgCu@MW-X (X is the co-sputtering time) in the subsequent discussion. The sequentially alternating sputtering substrates were denoted as Ag@Cu@MW-Y (Y is the number of alternating cycles). In this sputtering process, the targets and the moth wing templates were below and above, respectively. Moreover, the two targets were tilted to 45°, and the rotation velocity of the substrate was controlled at a uniform speed during sputtering to ensure its uniformity.

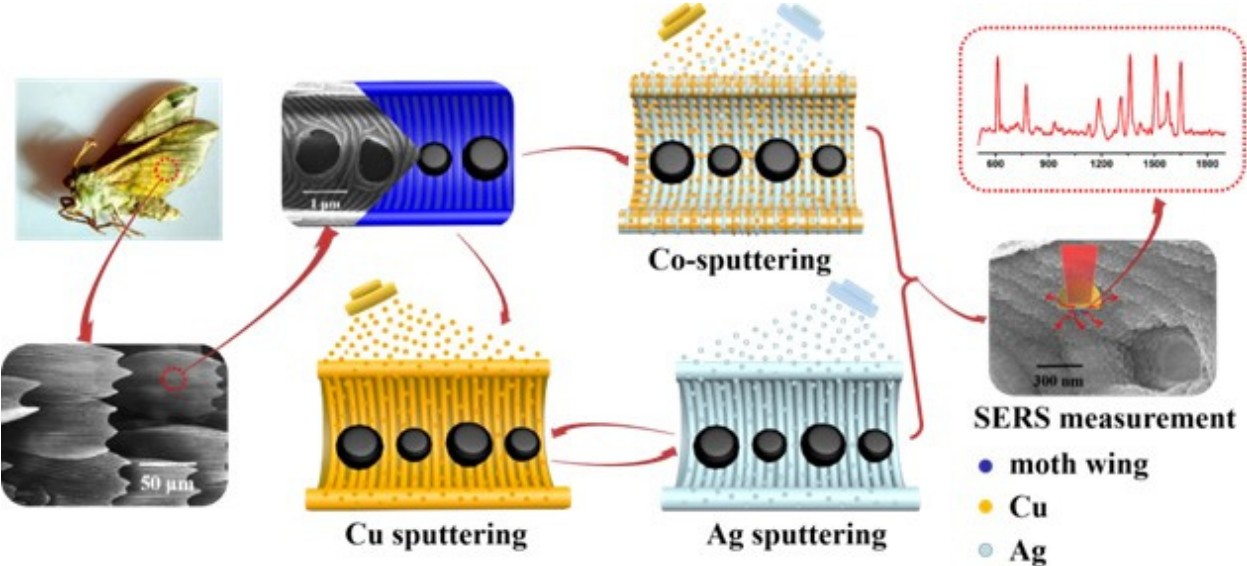

**Figure 1.** Schematic diagram of the fabrication process of the AgCu@MW and Ag@Cu@MW substrates by co-sputtering and sequentially alternating sputtering fashions, respectively.

### 2.3. SERS Measurement

In order to compare the Raman scattering intensity to pick out the optimum substrates, $10^{-5}$ M R6G aqueous solution was dropped onto the as-deposited AgCu@MW-X and Ag@Cu@MW-Y substrates. Subsequently, the R6G and MB aqueous solutions with $10^{-2}$ M were diluted to various concentrations (from $10^{-3}$ to $10^{-10}$ M). Then, these different concentrations solutions were dropped onto the optimum AgCu@MW and Ag@Cu@MW substrates. All of the Raman and SERS signals were collected using a 532-nm laser as the excitation source in this experiment, and the 10-s exposure time, 1-cm$^{-1}$ spectral resolution and the 50× aperture were used in all the Raman setup. The surface structure and laser polarization were calibrated using the image of optical microscopy under the optical settings of the Raman system before the measurements. During the whole experiment,

substrates taken out from the sputtering chamber were measured immediately to minimize the air exposure time and subsequent oxidation.

### 2.4. The Theoretical Analysis

The 3D finite-different time domain (FDTD) method was first proposed by Yee in 1966, which is a kind of method in the 3D mesh [35]. The Maxwell curl equation for an isotropic nondispersive medium in a passive region can be expressed as [36]:

$$\nabla \times \vec{E} = -\mu \frac{\partial \vec{H}}{\partial t} - \sigma_{\mathrm{m}} \vec{H} \tag{1}$$

$$\nabla \times \vec{H} = \varepsilon \frac{\partial \vec{E}}{\partial t} + \sigma_e \vec{E} \tag{2}$$

Among them, the $E$ and $H$ represented the electric field intensity and magnetic field strength, $\mu$ was the permeability, $\varepsilon$ was the representative dielectric constant and $\sigma_m$ and $\sigma_e$ represented equivalent magnetic resistance and electric conductivity, respectively. In FDTD simulations, the plane wave is usually replaced by a simulated wave packet, which is the incident on the nanostructure and propagates until the wave packet no longer interacts with the nanostructure. We use the modified Debye model to set the parameter values of silver and copper in the model [37]. In the process of calculating, we selected a 532-nm rectangular continuous excitation light source along the direction of the K incident to the basal surface, its polarization direction $E$ and observe the electric field intensity distribution in the plane.

## 3. Results and Discussion

### 3.1. Screening Substrate

Previous reports have shown that the differences in signal enhancement on the grating-like SERS-active substrates are mainly caused by the complex interactions among the following three factors: (i) strong localized [38,39] surface plasmon resonance of the template with a continuous grating structure, (ii) the additional surface plasmon resonance caused due to the surface roughness and (iii) the penetration depth of plasmons determined by the type and thickness of precious metal. Here, the nanogaps and distribution of the alloy were changed by co-sputtering and alternating the sputtering of Ag and Cu, and then, the high-density 'hot spots' were obtained. In order to compare the SERS performance of the as-prepared substrates, R6G is used as the probe molecule. Figure 2a,b shows the SERS spectra of $10^{-5}$ M R6G solution on the AgCu@MW substrates with different co-sputtering times and the Ag@Cu@MW substrates with different sequentially alternating sputtering layers. We found that AgCu@MW-20 and Ag@Cu@MW-3 have the higher enhancement in the co-sputtering and the alternating sputtering substrates. In addition, SERS intensities of R6G on the AgCu@MW-20 substrate were weaker than that on the Ag@Cu@MW-3 substrate, and the intensity was enhanced around eight times at the SERS peak of 1363 cm$^{-1}$. To estimate the SERS sensitivity of the screened-out AgCu@MW-20 and Ag@Cu@MW-3 substrates, the SERS spectra with various concentrations ($10^{-5}$ M to $10^{-10}$ M) of R6G were measured as shown in Figure 2c,d. We found that the Ag@Cu@MW-3 substrate ($10^{-10}$ M) possesses a higher sensitivity than the AgCu@MW-20 substrate ($10^{-8}$ M). Moreover, SERS peaks of R6G on all the substrates were identical in the position but different in intensity, which is consistent with previous reports and indicates the existence of R6G on all as-prepared SERS substrates [40].

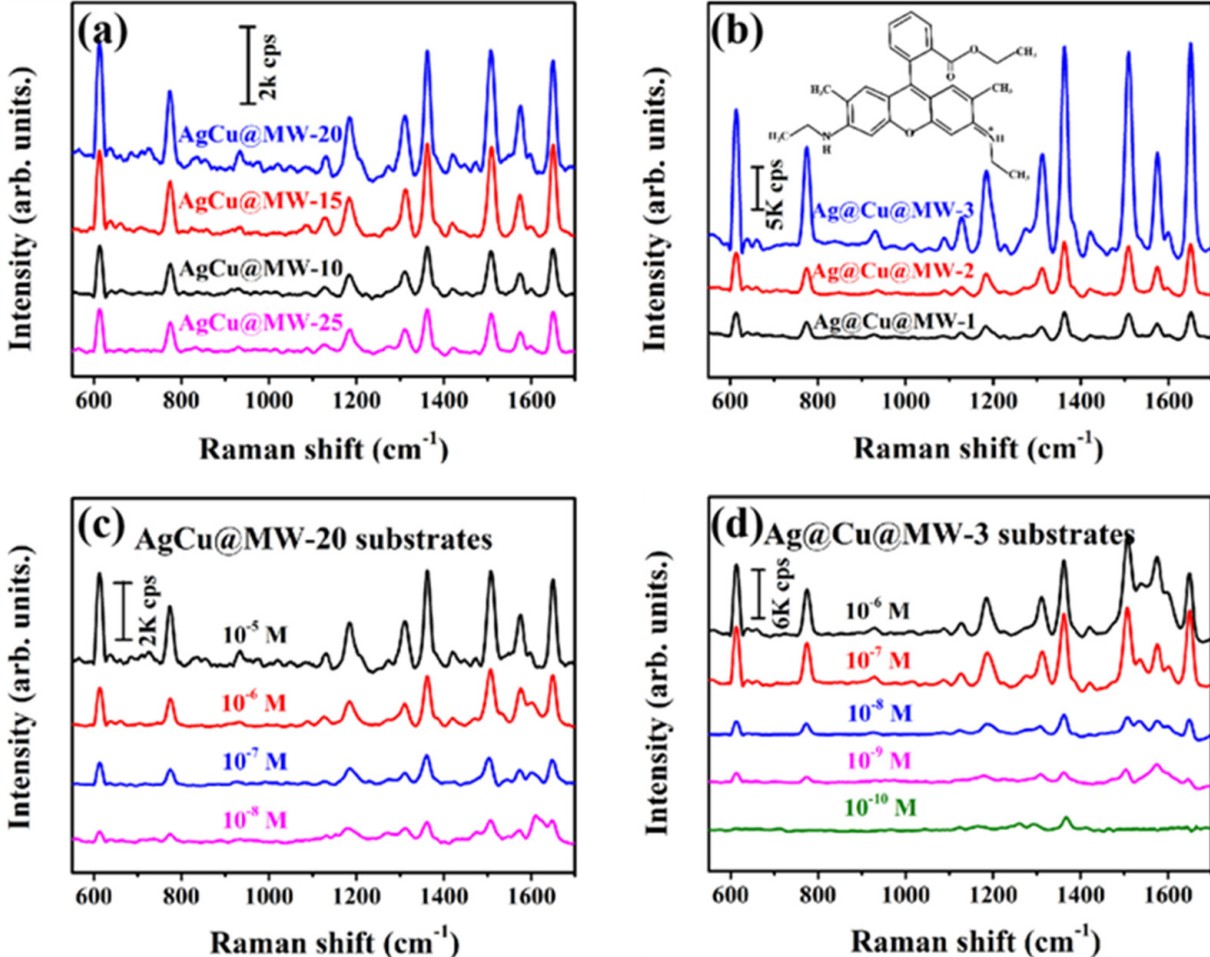

**Figure 2.** (**a**) SERS spectra of $10^{-5}$ M R6G solution on AgCu@MW substrates with different co-sputtering times. (**b**) SERS spectra of $10^{-5}$ M R6G solution on Ag@Cu@MW substrates with sequentially alternating sputtering. (**c,d**) SERS spectra collected from different concentrations of R6G solution absorbed on the AgCu@MW-20 and Ag@Cu@MW-3 substrates.

### 3.2. Surface Morphology and Chemical Composition

After the evolution of hundreds of millions of years, nature has created a large number of organisms with micro- and nanostructure surfaces in order to support the complex functions of the whole biological object. Moths are the maximum group in Lepidoptera, accounting for about nine-tenths of Lepidoptera species [41,42]. The surface of the moth wing templates is covered with closely and orderly domino-shaped scales on the micron scale, which are connected with the wing membranes through the scaly sacs at a certain angle. In addition, the surface of each scale is regularly distributed with approximately parallel longitudinal ribs and a groove structure, and the connected longitudinal ribs are connected by nano-parabolic ribs. The trapezoidal longitudinal ribs form different through the holes. The micromorphology of a moth wing with domino-shaped scales, the regularly distributed ribs and the groove structure are shown in Figure 3. The regular nanostructure of moth wings can provide a new idea for the creation of biomimetic and bioinspired surfaces, which may further promote the development of new composite materials. The top view FE-SEM images of the AgCu@MW substrates with different co-sputtering times are showed in Figure 3a–d. The images were obtained through moving the AgCu@MW SERS-active substrate onto the surface of a conductive carbon rubber strip of the SERS sample table using common tweezers. It is not difficult to find that the surface morphology of the AgCu@MW substrates has changed significantly; that is, the gap size decreased

between the ribs, and the surface roughness increased with the increase of the co-sputtering time of metal AgCu. When the co-sputtering time reached 25 min, the grating-like gap between the ribs almost disappeared. Therefore, the AgCu@MW-20 substrate obtained a high-density 'hot spot', which was in accordance with the above SERS measurement results in Figure 1a.

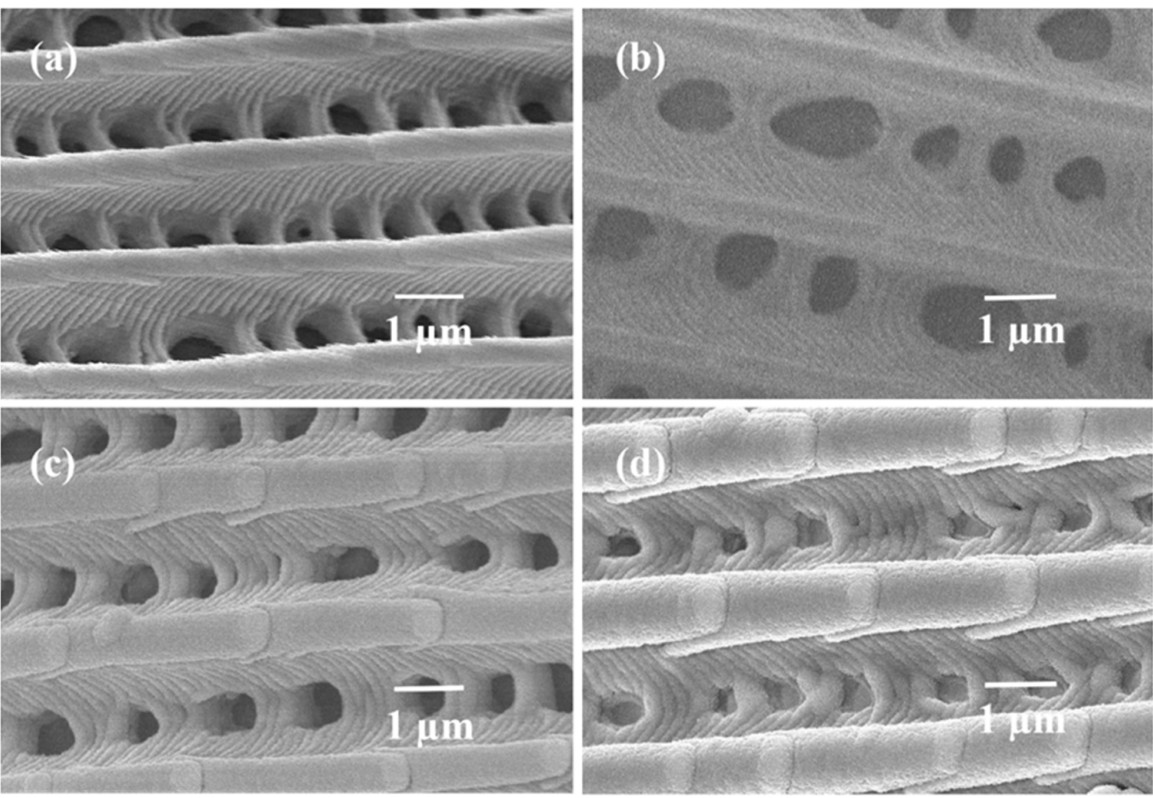

**Figure 3.** Top view FE-SEM images of the AgCu@MW substrates with different co-sputtering times: (**a**) AgCu@MW-10, (**b**) AgCu@MW-15, (**c**) AgCu@MW-20 and (**d**) AgCu@MW-25.

Figure 4a shows a FE-SEM image of the fabricated Ag@Cu@MW-3 substrate and its inset, i.e., the sequence of alternating sputtering metals (on the left side) and the enlarged Ag@Cu@MW-3 substrate (on the right side). According to previous studies, we know that Ag-coated Cu substrates not only have antioxidation properties but also have antibacterial properties [43]. Therefore, Ag nanoparticles were used to terminate the surface layer of the alternating sputtering substrates. The enlarged Ag@Cu@MW-3 image exhibited the nanogaps between the neighbouring decorated ribs, and the sizes were less than 10 nm. It could be also clearly seen that the longitudinal rib surface had many small protrusions and voids, which may be due to the larger atomic size and lower surface energy of Ag [44]. Two types of nanogaps will be possible to produce high-density 'hot spots' in the laser illumination area, and then, a strong SERS effect will emerge. In addition, the rough surface of the Ag@Cu@MW-3 substrate can take advantage to increase the adhesion area of the probe molecules. The element composition, weight and atomic distribution of the Ag@Cu@MW-3 substrate are analyzed by EDS in Figure 4b The EDS graph shows that the Ag@Cu@MW-3 substrate contains Ag and Cu elements. The element content from the inset shows that the weight ratio of Ag to Cu is 1:1, and the atomic ratio of Ag to Cu is 4:6. The SEM cross-section image was measured to characterize the thickness of the films sputtered sequentially. Figure 4c shows the cross-section image of the Ag@Cu@MW-3 substrate. Distinct layered structures were revealed, which were due to the sputtering of the Cu and Ag targets alternately on three occasions. The red and blue lines represented the heights of the Ag and Cu layers, respectively. According to the particle size measurements, the

average thickness of Ag was about 134 nm, while the average thickness of Cu was around 54 nm. The UV–Vis absorption spectra of Ag@Cu@MW-1 (black curve), Ag@Cu@MW-2 (red curve) and Ag@Cu@MW-3 (blue curve) in Figure 4d are measured to show their SPR nature. It is obvious that the main absorption peaks of these three substrates were near 318 nm, which is due to the fact that the outermost layer of the substrate was thin-film Ag. However, they were different at the absorption peak near 361 nm, which was highlighted in the blue area. This originated from the different numbers of alternating cycles of Ag and Cu targets. The more sputtering layers, the more gradual the absorption. This means that the SPR of the substrate will change slightly in response to different structures.

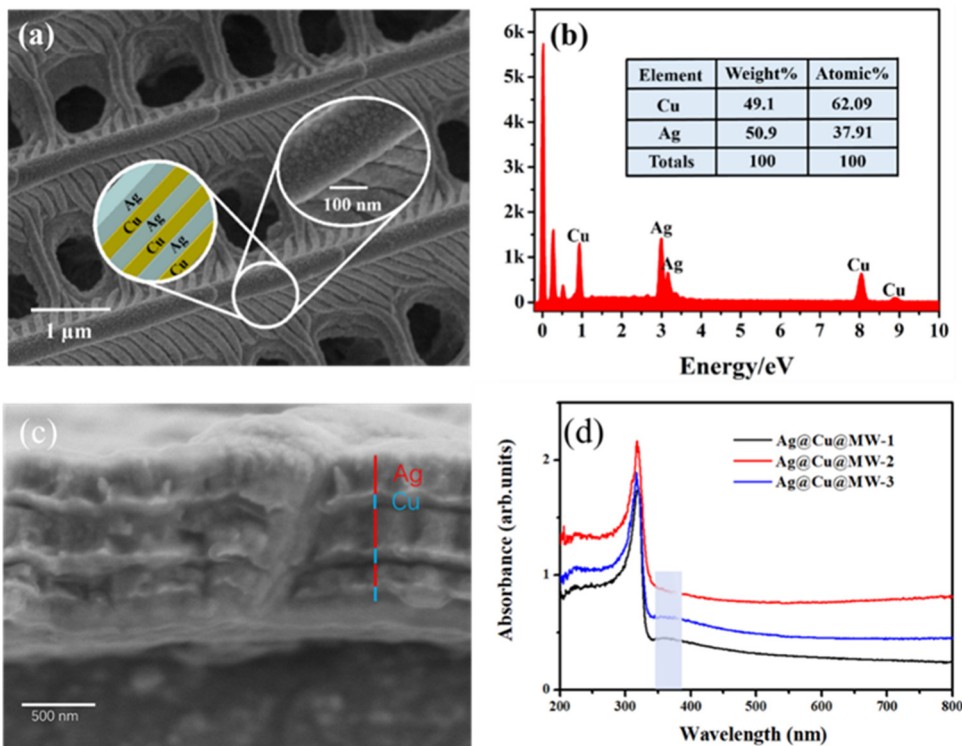

**Figure 4.** (**a**) Top view FE-SEM image of the Ag@Cu@MW-3 substrate. (**b**) EDS graph of the Ag@Cu@MW-3 substrate. (**c**) SEM cross-section image of the Ag@Cu@MW-3 substrate. (**d**) UV–Vis absorption spectra of Ag@Cu@MW-1, Ag@Cu@MW-2 and Ag@Cu@MW-3.

*3.3. Comparison of Reproducibility and Homogeneity of AgCu@MW-20 and Ag@Cu@MW-3 Substrates*

MB, as a dye commonly used in printing cotton, wood and silk, can not only cause eye burns but also cause permanent damage to humans and animals if it is serious [45,46]. Therefore, it is very meaningful to use an efficient, fast and sensitive method to detect MB dyes in wastewater. The reproducibility of SERS signals on a prepared substrate is a vital standard for conventional SERS applications. As shown in Figure 5a,b, a series of SERS spectra of $10^{-5}$ M MB at 36 randomly selected points from the AgCu@MW-20 and Ag@Cu@MW-3 substrates were collected to examine the reproducibility of substrate-to-substrate, and the results illustrated that the peak locations and intensities of different spectra at all of the AgCu@MW-20 and Ag@Cu@MW-3 substrates were almost identical. In addition, the SERS contour plots in Figure 5c,d further show that the position and intensity of SERS spectra are stable relative to the position variations of laser spot irradiation. Therefore, both the AgCu@MW-20 and Ag@Cu@MW-3 substrates have wonderful substrate-to-substrate reproducibility. Furthermore, the point-to-point homogeneity of the AgCu@MW-20 and Ag@Cu@MW-3 substrates was also evaluated by using $10^{-5}$ M MB solution at a characteristic peak of 1622 cm$^{-1}$ recorded from 36 randomly selected points. The SERS intensity (Figure 6a,b) and corresponding mapping images (Figure 6c,d) were

shown in Figure 6. All 36 points of peak height in Figure 6a,b were basically identical. Moreover, the values of RSD of the AgCu@MW-20 and Ag@Cu@MW-3 substrates were 16.63% and 10.15%, respectively. It can be seen that the point-to-point homogeneity of the Ag@Cu@MW-3 substrate was much higher than that of the AgCu@MW-20 substrate.

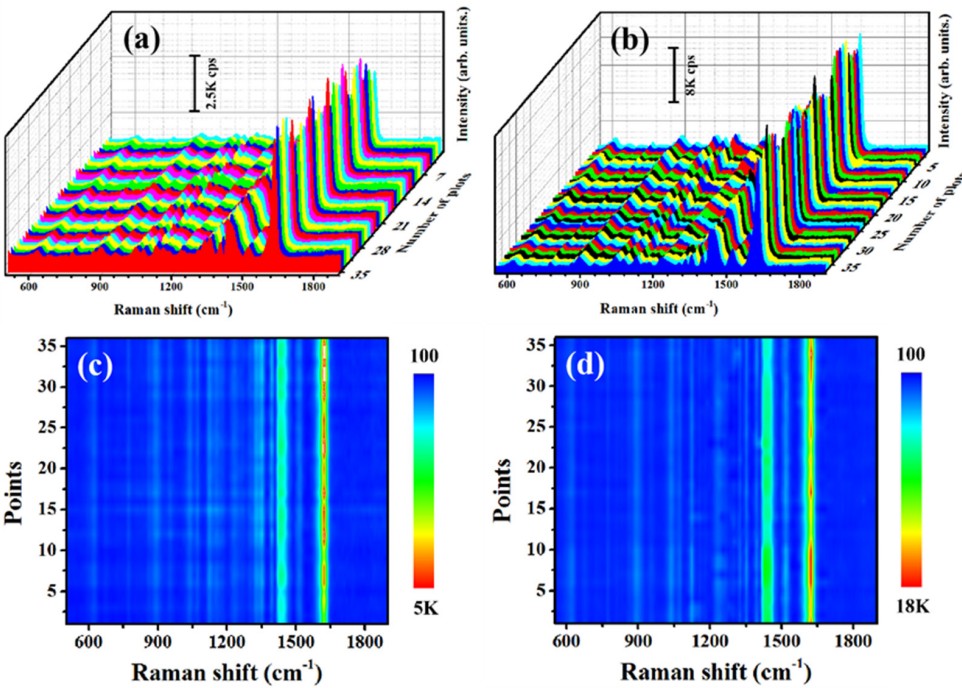

**Figure 5.** (**a**,**b**) SERS spectra of the $10^{-5}$ M MB solution recorded from 36 randomly selected points from the AgCu@MW-20 and Ag@Cu@MW-3 substrates. (**c**,**d**) The SERS contour plots of $10^{-5}$ M MB collected from 36 randomly selected points from the AgCu@MW-20 and Ag@Cu@MW-3 substrates.

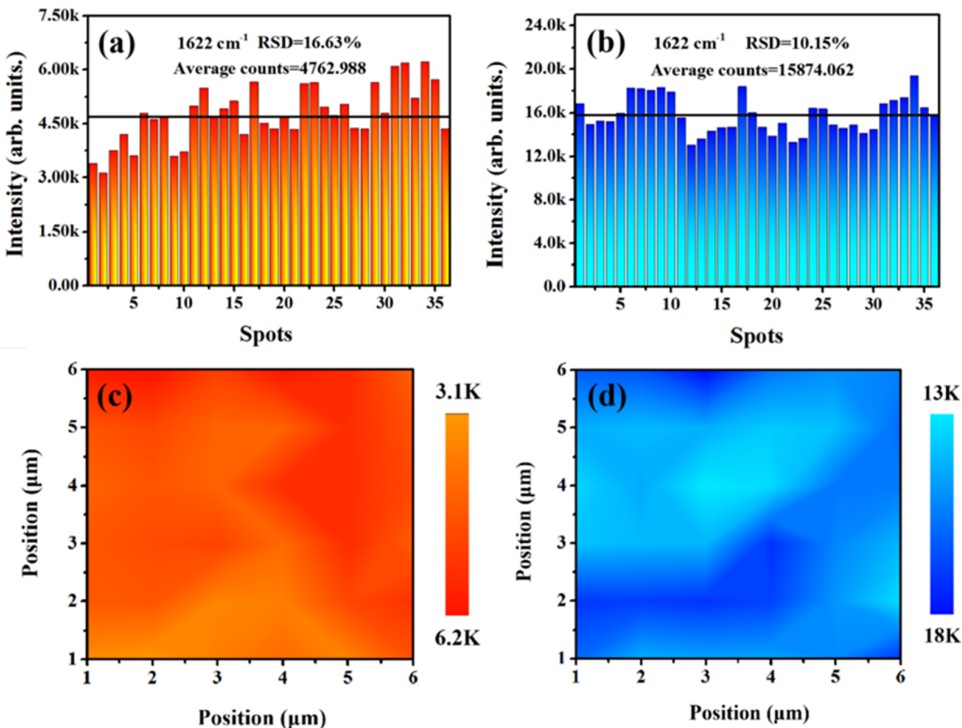

**Figure 6.** (**a**,**b**) SERS intensity of $10^{-5}$ M MB at a characteristic peak of 1622 cm$^{-1}$ recorded from

36 randomly selected points from the AgCu@MW-20 and Ag@Cu@MW-3 substrates. (**c**,**d**) SERS mapping image (6 × 6 μm², 1-μm step size) corresponding to the SERS intensity of the above 36 points.

### 3.4. Three-Dimensional Finite-Difference Time-Domain Simulation

To further illustrate this enhancement effect, the 3D-FDTD is applied to simulate the locally electric field distribution of all AgCu@MW and Ag@Cu@MW SERS-active substrates based on the morphology from FE-SEM images in Figures 3 and 4. In the simulation process, the enhancement factor (EF) of the SERS-active substrates can be approximated to the fourth power of $(E/E_0)$, where $E$ and $E_0$ represent the electric field amplitude and incident light amplitude, respectively [47]. It needs pointing out that the incident electromagnetic field $E_0$ was set to 1. Figure 7e–h reveals the spatial distribution of the electric field intensities of the Figure 7a–d 3D-FDTD models. Among them, a rectangular continuous incident wave vector of a 532-nm laser was along the $K$ direction, and the polarization vector was along the $E$ direction. As is known, the smaller gap SERS substrate can significantly increase the plasmonic coupling and SERS enhancement factor [11]. According to the obvious color contrast in Figure 7e–h, the simulation results show that the AgCu@MW-20 substrate has a more prominent SERS effect, and the calculated EF value was $9.9957 \times 10^5$. This powerful SERS enhancement of the AgCu@MW-20 substrate can be attributed to the existence of a large number of sub-10 nanogaps of 'hot spot II;' in the unit illumination area. In addition, the spatial distribution of the electric field intensities of the Ag@Cu@MW substrates is illustrated in Figure 8d–f. The SERS enhancements of the Ag@Cu@MW-1 ($1.376 \times 10^6$), Ag@Cu@MW-2 ($1.109 \times 10^6$) and Ag@Cu@MW-3 ($1.307 \times 10^6$) substrates show no obvious changes, which may be attributed to the unchanged nanogap among the Ag@Cu@MW substrates.

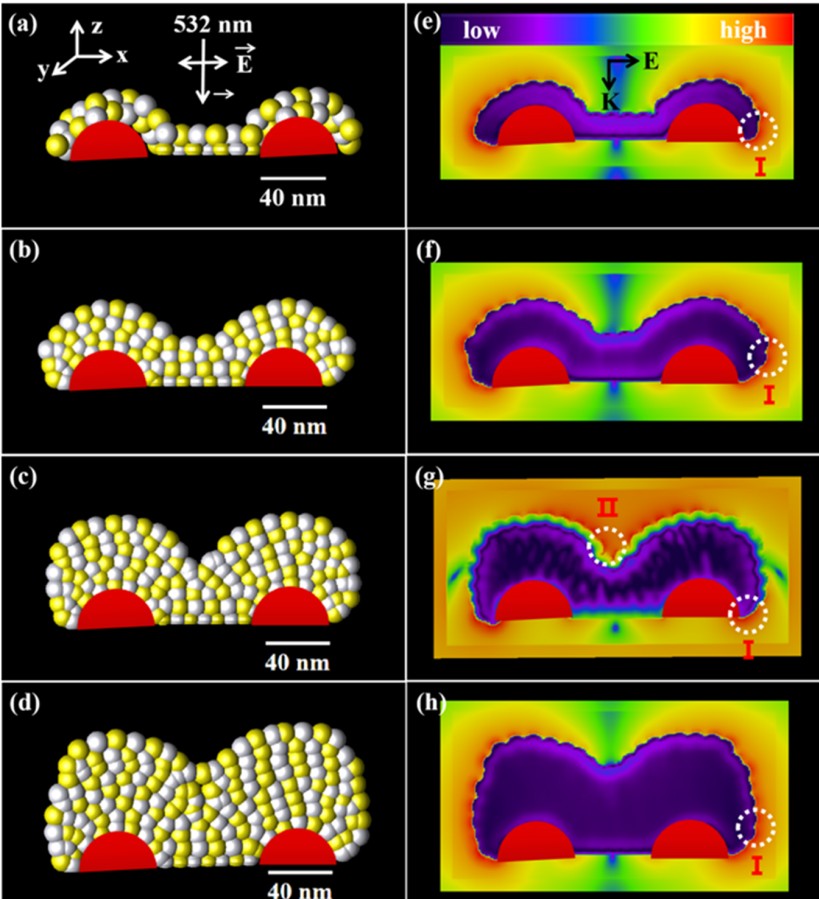

**Figure 7.** (**a**–**d**) Three-dimensional FDTD models of AgCu@MW substrates with different co-sputtering

times. (**e–h**) The electric near-field intensity maps for the x–z plane around the corresponding substrates in (**a–d**).

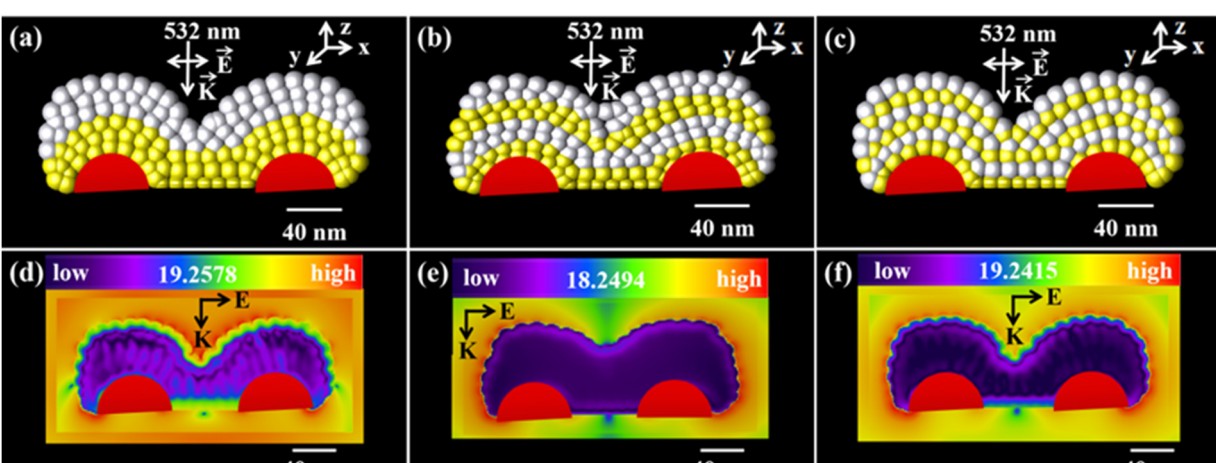

**Figure 8.** (**a–c**) Three-dimensional FDTD models of Ag@Cu@MW substrates with different alternating sputtering layers. (**d–f**) The electric near-field intensities maps for the x–z plane around the corresponding substrates in (**a–c**).

## 4. Conclusions

In conclusion, we adopted two different combinations of metal Ag and Cu to prepare large-scale and cost-effective SERS-active substrates by decorating precious metals Ag and Cu onto MWs of 3D grating-like nanoarrays. One is to effectively adjust the gap size between the ribs by changing the time of the co-sputtering of AgCu. The other is to fabricate a layered structure by alternating sputtering under the conditions of the above optimum gap size, alternating a sputter-deposited layered Ag@Cu@MW-3 substrate with a higher SERS performance as compared with the co-sputtered AgCu@MW-20 substrate. These include the Raman intensity of R6G molecules at 1363 cm$^{-1}$ as about eight times higher than that of the optimum co-sputtered substrate; the sensitivity can be increased by two orders of magnitude compared with the optimum co-sputtered substrate; the RSD value can be lower than the optimum co-sputtered substrate. Moreover, comparison research of the SERS performance between co-sputtering and sequential sputtering substrates can provide insights into the future design and application of SERS substrates.

**Author Contributions:** Data curation, X.S., X.Y. and N.L.; Investigation, X.S., X.Y. and N.L.; Writing—original draft, X.S., X.Y. and N.L.; Validation, Visualization, X.S., X.Y. and N.L.; Conceptualization, L.S. and M.W.; Data curation, L.S. and M.W.; Supervision, L.S. and M.W. All authors have read and agreed to the published version of the manuscript.

**Funding:** This work was supported by the National Natural Science Foundation of China (51771162 and 21872119) and Special Project of Scientific and Technological Innovation Ability of College and Middle School Students in Hebei Province (1120009).

**Institutional Review Board Statement:** Not applicable.

**Informed Consent Statement:** Not applicable.

**Data Availability Statement:** Data sharing is not applicable to this article.

**Conflicts of Interest:** The authors declare no conflict of interest.

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
