# Peer review of "Study on the Performance of Ag-Cu Bimetal SERS Substrate"

_coatings, doi:10.3390/coatings12101457_

Round 1
Reviewer 1 Report
In this paper, the authors have presented the study of SERS Properties with Decorated Different Combina-2 tion AgCu Metal onto Grating-Like Moth Wing Scale Array. The results indicated that the stratified structure 20 Ag@Cu@MW-3 substrate have high SERS performance, justifying that the strategy can be stretched to the preparation of other potential substrates with high optical properties for other applications as well. The work is interesting and the data obtained is well-formulated and discussed. After minor technical corrections, it can be considered for publication in the journal.
Author Response
Thank you for your professional advice. We have made technical modifications.
Reviewer 2 Report
The work by Son et al. (Study of SERS Properties with Decorated Different Combination AgCu Metal onto Grating-Like Moth Wing Scale Array) reports theemployment of bimetallic nanostructures of Ag and Cu prepared by changing the metal content ratio. In the study, they used the scale of moth wings (MW) with 3D grating-like uniform nanoarrays as bioscaffold. Co-sputtered with different time and sputtered with sequentially alternating metals approcahes were used for the fabrication of the SERS-active substrate. The SERS activity of the platforms were by using probe molecules R6G and MB.
The study seems to be interesting and provides valuable information to this research field. However, additional studies must be perrformed to improve the quality of the report. Therefore, I recommend the publication of the report after some critical points given below are addressed.
Please reconsider the title of the report. I think better options are available for the study.
Please use the abbreviations (AgCu@MW and Ag@Cu@MW) when it is first used in the Abstract.
Line 33 and 43. The abbreviation was defined twice. Please fix this issue.
There is no data for the FDTD analsis in the Experimental section. Please provide how this analysis was performed.
Please provide the thickness of each layer of Ag or Cu during a deposition for each SERS platform.
The authors must provide UV-vis absorption spectra of SERS platforms for the evaluation of their SPR nature. This data would be informative to clarify the SERS activity.
For Ag@Cu@MW-3 in Figure 4, depth-profile analysis through XPS is required to show the layer-by-layer structure of Ag and Cu.
For Figure 4, the upper layer of the substrate (Ag or Cu) is the main contributor to SERS effect. Ag and Cu would create different SERS performences due to their different SPR characteristics. Unfortunately, this issue was neglected by the authors.
I noticed many grammatical errors. Please fix these issues.
Author Response
Thanks

Reviewer 3 Report
Review of the manuscript "Study of SERS Properties with Decorated Different Combination AgCu Metal onto Grating-Like Moth Wing Scale Array". It deals about the experimental analysis based on SERS technique of coated substrates.
The manuscript is clear and well arranged.
I can just suggest to improve english and revise references list according to the journal requirements. Also in the text references are not correctly called.
After that the manuscript can be accepted.
Author Response
Thanks

Reviewer 4 Report
The author adopted a simple approach to fabricating bimetallic nanostructured substrates by varying Ag and Cu combinations and used Moth Wing as a bioscaffold. The analysis of surface-enhanced Raman spectroscopy was carried out systematically and presented in a convincing way.
My suggestion is to specify the application focus in the conclusion section. This paper can be accepted, but certainly, language /grammatical errors have to be corrected before publishing it. The authors narrated certain sections in the past tense and a few sections in the present tense. Authors should consider checking the manuscript with a native English speaker if possible.
Author Response
Thanks

Round 2
Reviewer 2 Report
I recommend the publications of the report in the presented form.